# A structured evaluation of genome-scale constraint-based modeling tools for microbial consortia

**William T. Scott Jr.**[1,2☯*], **Sara Benito-Vaquerizo**[1☯], **Johannes Zimmermann**[3], **Djordje Bajić**[4], **Almut Heinken**[5], **Maria Suarez-Diez**[1], **Peter J. Schaap**[1,2]

**1** Laboratory of Systems and Synthetic Biology, Wageningen University & Research, Wageningen, the Netherlands, **2** UNLOCK, Wageningen University & Research and Delft University of Technology, Wageningen, the Netherlands, **3** Christian-Albrechts-University Kiel, Institute of Experimental Medicine, Research Group Medical Systems Biology, Kiel, Germany, **4** Department of Biotechnology, Delft University of Technology, Delft, the Netherlands, **5** Inserm U1256 Laboratoire nGERE, Université de Lorraine, Nancy, France

☯ These authors contributed equally to this work.
\* william.scott@wur.nl

**Data Availability Statement:** The code used to produce our reported results and raw data can be downloaded from: https://doi.org/10.5281/zenodo.

## Abstract

Harnessing the power of microbial consortia is integral to a diverse range of sectors, from healthcare to biotechnology to environmental remediation. To fully realize this potential, it is critical to understand the mechanisms behind the interactions that structure microbial consortia and determine their functions. Constraint-based reconstruction and analysis (COBRA) approaches, employing genome-scale metabolic models (GEMs), have emerged as the state-of-the-art tool to simulate the behavior of microbial communities from their constituent genomes. In the last decade, many tools have been developed that use COBRA approaches to simulate multi-species consortia, under either steady-state, dynamic, or spatiotemporally varying scenarios. Yet, these tools have not been systematically evaluated regarding their software quality, most suitable application, and predictive power. Hence, it is uncertain which tools users should apply to their system and what are the most urgent directions that developers should take in the future to improve existing capacities. This study conducted a systematic evaluation of COBRA-based tools for microbial communities using datasets from two-member communities as test cases. First, we performed a qualitative assessment in which we evaluated 24 published tools based on a list of FAIR (Findability, Accessibility, Interoperability, and Reusability) features essential for software quality. Next, we quantitatively tested the predictions in a subset of 14 of these tools against experimental data from three different case studies: a) syngas fermentation by *C. autoethanogenum* and *C. kluyveri* for the static tools, b) glucose/xylose fermentation with engineered *E. coli* and *S. cerevisiae* for the dynamic tools, and c) a Petri dish of *E. coli* and *S. enterica* for tools incorporating spatiotemporal variation. Our results show varying performance levels of the best qualitatively assessed tools when examining the different categories of tools. The differences in the mathematical formulation of the approaches and their relation to the results were also discussed.

8074832 and https://gitlab.com/wurssb/Modelling/modelingtools_microbial_consortia.

**Funding:** This work was supported by the Dutch Research Council (Nederlandse Organisatie voor Wetenschappelijk Onderzoek (NWO)), and Wageningen University and Research through the UNLOCK initiative (NWO: 184.035.007 to P.J.S. and W.T.S.J.). This work was also supported by NWO under the Programme 'Closed Cycles' (Project NR. ALWGK.2016.029 to M.S-D) and the Netherlands Ministry of Education, Culture and Science under the Gravitation (Grant NR. 024.002.002 to S.B-V.). The funders had no role in study design, data collection and analysis, decision to publish, or preparation of the manuscript.

**Competing interests:** I have read the journal's policy and the authors of this manuscript have the following competing interests: AH is one of the developers of the Microbiome Toolbox. JZ is one of the developers of BacArena. DB is one of the developers of COMETS.

Ultimately, we provide recommendations for refining future GEM microbial modeling tools.

## Author summary

Constraint-based modeling employing genome-scale reconstructions of microbial species has become one of the most successful approaches for studying, analyzing, and engineering microbial consortia. Over the past decade, many constraint-based modeling tools have been published to examine an immense variety of microbial consortia spanning from the application areas of bioremediation to food and health biotechnology. However, new potential users lack an overview of the quality and performance of existing metabolic modeling tools that would guide their choice. To tackle this issue, we examined 24 tools for genome-scale metabolic modeling of microbial consortia. After an initial qualitative screening, we quantitatively evaluated 14 adequate tools against published experimental data that included different organisms and conditions. We conducted simulations and evaluated model features such as predictive accuracy, computational time, and tractability in capturing critical physiological properties. We found that, generally, more up-to-date, accessible, and documented tools were superior in many important aspects of model quality and performance. Although, in some cases, we observed tradeoffs in older, less elaborate tools that can be more accurate or flexible. This work has broad implications to help researchers navigate the most suitable tools, and suggests to developers opportunities for improvement of the currently existing capabilities for metabolic modeling of multi-species microbial consortia.

## Introduction

The goods and services provided by microbial consortia have long been harnessed in biotechnology. Recent years have seen a soar in the use of multi-species microbial consortia beyond their traditional application in the production of food and beverages [1, 2]. For example, microbial consortia are increasingly applied in the production of commodity chemicals [3, 4], pharmaceuticals [5], or biofuels [6–8]; the valorization of waste and emissions [9–11], the improvement of sustainable agriculture systems [12–14], and applications in health [15] or bioremediation [16–18]. Besides the obvious advantage over established industrial processes in terms of sustainability, multi-species consortia offer a number of advantages compared to monocultures. These include the reduction of metabolic burden (a major issue) through division of labor [19–21], an enhanced substrate versatility, and an increased robustness to fluctuating environments [22, 23]. In consequence, synthetic ecology—the rational engineering of multi-species microbial consortia—is emerging as a new frontier in biotechnology and biomedicine. In order to advance this frontier, it is imperative to build predictive models that will allow us to design and control the composition and function of microbial communities [24, 25].

Constrained-based metabolic modeling (CBM) is a powerful computational approach that mechanistically predicts microbial metabolic traits from genomes. In the past decades, this method has proven invaluable as a guide for microbial experimental design and for elucidating metabolic engineering strategies [26, 27]. Because metabolic traits are also a central determinant of ecological interactions in microbes (e.g. competition for resources or metabolite

sharing), CBM also holds great promise as a predictive and an engineering tool in synthetic ecology [4]. In a nutshell, CBM use genome-scale metabolic models (GEMs), a mathematical representation of the metabolic network encoded in an organism's genome, to simulate metabolic fluxes in a given environment [28]. In the case of single organisms, one of the most popular methods is Flux Balance Analysis (FBA) [29]. FBA optimizes a predefined objective function (e.g., biomass production) and assumes steady-state exponential growth (balanced growth). Dynamic-FBA (dFBA) -an extension of FBA- is applied to represent non-continuous operations, such as batch or fed-batch reactors, by incorporating differential equations that describe the rate of change of the extracellular fluxes and the mass balances of the reactor [30, 31]. dFBA assumes a quasi-stationary state where internal dynamics are supposed to be much faster than external changes of the medium [32]. Going a step further, another general methodology has been developed to model microbial systems in which the extracellular environment varies spatially and temporally [33–35]. This approach is known as spatiotemporal FBA. A spatiotemporal FBA framework typically consists of partial differential equations (PDEs) conveyed in terms of time and spatial coordinates as independent variables [34]. The PDEs mainly characterize extracellular mass balance equations for biomass, metabolite, and potentially other chemical species concentrations. They account for the transport mechanisms that cause spatial disparities, including metabolite diffusion and liquid/gas phase convection. Some spatiotemporal FBA approaches choose to represent microbial biomass as individuals or agents (IBM) [36–38]. In contrast, others opt to represent microbial biomass at the population level (PLM) [33, 35].

With the continuous development of novel synthetic microbial consortia, considerable efforts have been made to extend CBM to microbial communities [39–43]. In the past decades, many tools have become available with the aim of studying microbial interactions in the gut microbiome [44–46] or simulating the growth of microbial consortia in continuous and non-continuous environments. The increasing availability of these tools makes the selection process difficult for the user [47]. A key feature when selecting a tool is to verify whether it follows the FAIR (Findable, Accessible, Interoperable, and Reusable) principles [48]. In particular, adhering to the software FAIR guiding principles is best as they assure quality research maintenance and reproducibility [49, 50]. As such, the findability of a tool is based on the capacity of the metadata and software to be easily found by both humans and computers. Accessibility pertains to the ease of knowledge available at which a software tool can be accessed, possibly including authentication and authorization. Interoperability refers to the ability to communicate with other software via exchanged data (or metadata). The reusability of tools is related to how well-described (by metadata) and appropriately structured the software is so that outputs/results can be replicated, combined, reinterpreted, reimplemented, and/or used in different settings. Many studies have reviewed the state of the art of steady-state, dynamic, or spatiotemporal tools and followed qualitative assessments [39–43, 51–54]. However, no study has yet reviewed these tools quantitatively. Thus, a systematic evaluation of the latter element would be highly beneficial for the users and developers in the field. In this work, we have followed an extensive qualitative assessment to evaluate FAIR principles of available tools and a quantitative assessment to evaluate the performance of a subset of tools reproducing available experimental data of two-species communities. The following case studies were selected to quantitatively evaluate the tools: i) syngas fermentation by *Clostridium autoethanogenum* and *Clostridium kluyveri* for the static tools, b) xylose and glucose mixture fermentation with engineered *Escherichia coli* and *Saccharomyces cerevisiae* for the dynamic tools, and c) a Petri dish of *E. coli* and *Salmonella enterica* for the spatiotemporal tools. In particular, we tested tools that use CBM and GEMs; based on steady-state, dynamic, or spatiotemporal conditions and suitable to model synthetic microbial communities of two species. Ultimately, we made

recommendations for the best modeling tools to use based on qualitative and quantitative performance outcomes and for the future development and improvement of the tools.

## Overview of constrained-based modeling tools/approaches

Most of the community modeling tools available are based on steady-state, dynamic, and spatiotemporal conditions, describing continuous cultures, non-continuous cultures, and complex liquid and/or solid systems, for instance, a preferred environment of a Petri dish, respectively (Fig 1). Steady-state approaches are suitable to describe growth in chemostats or continuous stir batch reactor (CSBR) systems. These tools require a single GEM of individual species and a community GEM, often generated by the tool. Extracellular metabolites and reactions of single species models must be defined by the user in the same namespace (unified identifier) upon constructing the community model. Most tools require the definition of medium composition, and some require either the relative abundance (microbial composition) or the growth rate of individual species. The community growth is often defined as the objective function, and/or the species growth. The objective function is maximized under the specified constraints by computing the metabolic fluxes, microbial composition, or species growth rate, thus allowing a solution where microbial interactions can be inferred.

Tools based on dynamic approaches are suitable to describe non-continuous systems, such as batch serum bottles, batch or fed-batch reactors as well as some continuous systems, e.g., temporal dynamics of recovery from a perturbation in a CSBR. GEMs of species in the community are provided separately as inputs. All the tools require the medium composition in the form of initial concentration and substrate uptake rates and the kinetic parameters as inputs.

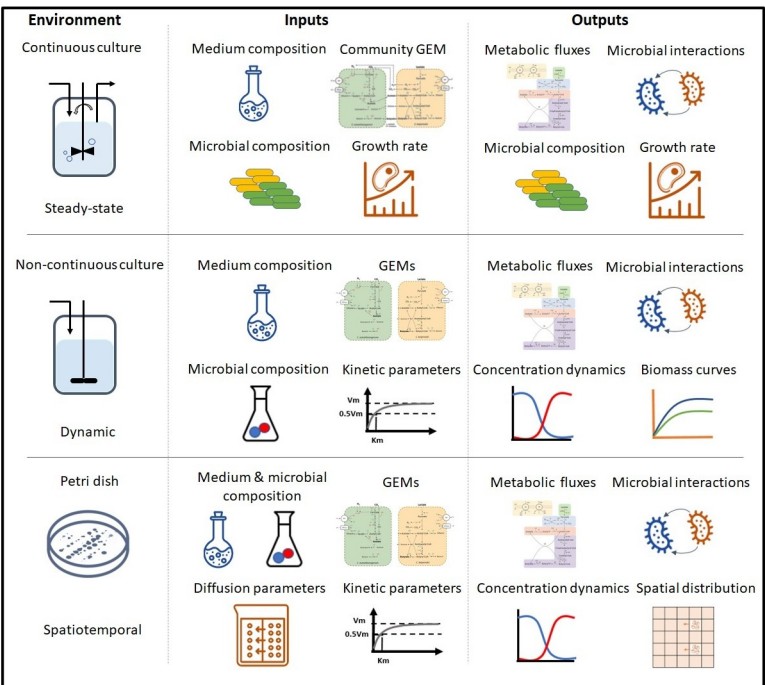

**Fig 1. Overview of steady-state, dynamic, and spatiotemporal tools to model microbial communities.** The division among steady-state, dynamic, and spatiotemporal tools is not firm since dynamic tools can be used to describe steady-state systems. Likewise, spatiotemporal tools can be used to describe dynamic and steady-state systems. However, most tools were specifically designed to be used for the highest dimensional cases.

The kinetic parameters are normally based on Michaelis-Menten-like kinetics, and thus, Michaelis-Menten constant ($K_m$) and maximum uptake rate of substrates ($q_{Si,m}$) are required parameters. After optimization of individual species' growth rate, we can obtain information on cross-feeding metabolites, concentration dynamics of substrates, products and biomass, and metabolic fluxes.

Finally, the spatiotemporal tools aim to describe 2D dimensional surface environments such as mimicking simple solid-state Petri dish environments. In addition to the required inputs described for dynamic tools, these tools need information on the diffusion parameters. The diffusion parameters consist of diffusion coefficients for biomass and metabolites. The main output in spatiotemporal models is the spatial distribution of extracellular metabolites, biomass of the different species, as well as growth and uptake rates, at any given time point. This information can be then used to resolve space-dependent ecological interactions. For instance, we can observe how growth and competition for substrates occurs at the border of a colony, but not in the interior.

In this study, we evaluated a total of twenty-four tools/approaches based on steady-state (9) [42, 44–46, 55–60], dynamic (8) [61–68] and spatio-temporal (7) [33, 35–38, 69–71] methods according to their usability to model microbial communities using GEMs. A description of the tools/approaches is found in the S1 Text.

## Results

Both a qualitative and a quantitative assessment were performed to evaluate the modeling tools and approaches. The assessment workflow began by first constructing a list of 16 essential features for quality constraint-based GEM modeling of microbial communities. Each tool was rated according to performance on these qualitative metrics ranging from 1 for inadequate (Red) to 5 for excellent (Blue). The qualitative features are strongly related to FAIR Guiding Principles for research software which include aspects such as software availability, user support, traceability, interoperability, etc [50]. These 16 features were evaluated from the perspective of a fairly experienced user of COBRA methods and tools. Some features can be subjective. Thus, a description of the criteria followed to evaluate each qualitative metric was described in an evaluation rubric (S2 Table). More specifically, to evaluate the numerical stability and reproducibility of the tools, we relied on existing literature, tool architecture, and/or their performance in reproducing the available experimental studies chosen for the quantitative assessment. Not all features could be examined since some tools were not readily available, there was no tool developed for those approaches, or there was a lack of related information in the literature. These features were marked as 'Not applicable' (NA; grey square).

From the qualitative examination of the tools, a subset of tools/approaches were quantitatively evaluated for their potential to directly model microbial consortia under various conditions. For this subsequent evaluation, we needed data pertaining to substrate uptake rates, biomass composition and growth rates for static tools, substrate and product concentrations and biomass concentrations over time for dynamic tools, and media concentrations and spatial distribution of metabolites for spatiotemporal tools. However, these types of raw data are rarely available along with their metadata. Therefore, only a few suitable candidate datasets were applicable to the surveyed modeling scenarios. We selected three datasets (Diender et al. [72], Hanly and Henson [73], and Harcombe et al. [33]), one dataset for each category of modeling tools to use as case studies to validate the predictive capabilities of the tools. Every case study represents a consortium of two species. To have an unbiased comparison, tools were not modified or augmented to revise their functionalities. Some tools were also available,

but they were not quantitatively assessed since they were too specific to a particular application and not designed for general application.

In these evaluations, two main points were addressed: the quality of the tool for modeling microbial communities, and the predictability of the community behavior by the tool when presented with data from simple test cases.

## Qualitative assessment—Static tools/approaches

Fig 2 shows the evaluation of the described features in every tool/approach based on the rubric (S2 Table). Joint FBA was not assessed since it is the oldest approach and did not include important parameters such as the relative abundance of species. cFBA, RedCom, and NECom are approaches, and therefore, some features were not evaluated (grey squares) since there is no tool developed specifically for those approaches. The tools/approaches that best meet FAIR principles are MICOM, MMT, SteadyCom, and cFBA (see scores in S1 Fig). All tools/approaches are accessible for external use except CASINO and RedCom. OptCom is freely available for academic users upon request.

OptCom did not undergo updates throughout the years. SteadyCom is integrated as part of COBRA Toolbox [74] and has had some updates from the original publication. Small issues are fixed from time to time in SteadyCom. However, changes are not documented clearly. MMT and MICOM are routinely updated, and developers fix issues. Besides, all the modifications applied in every update of MICOM are described.

OptCom does not have community repository support, and developers only provide a contact person. SteadyCom and MMT offer good community support within COBRA Toolbox.

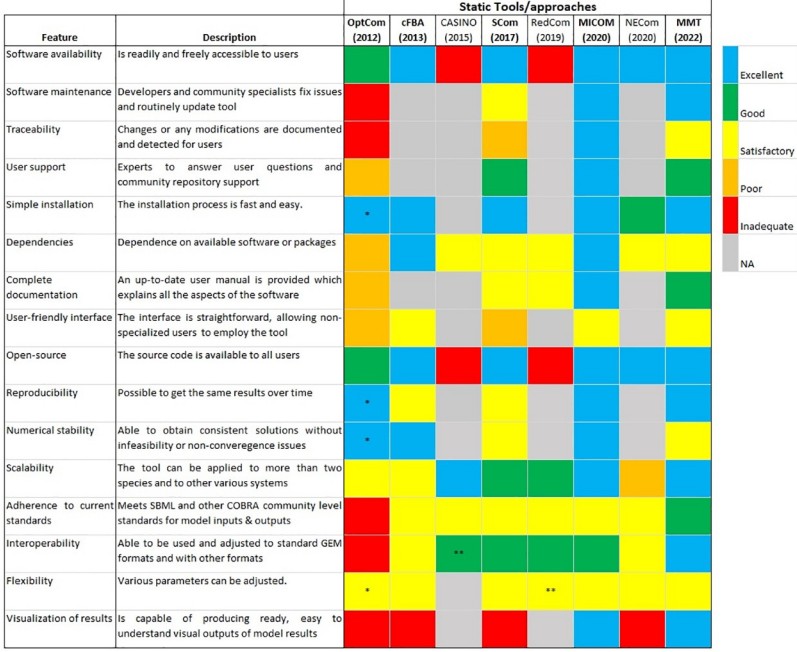

**Fig 2. Qualitative assessment of the static tools/approaches.** The colored squares indicate the evaluation of the specified feature in every tool/approach. The color scale (upper right) goes from excellent (blue) to inadequate (red). When a feature does not apply to the specified tool/approach or the feature was not evaluated, it is indicated as NA (Not applicable; grey). The metrics contained in the figure were inspired by [47]. Colored squares with '\*' indicate features of OptCom evaluated using the OptCom function from MICOM. Colored squares with '\*\*' indicate that the evaluation is an assumption based on the given information. Scom: SteadyCom; MMT: Microbiome Modelling Toolbox. The tools/approaches are ordered by the year of the latest publication (e.g., MMT).

MICOM has excellent user support since it has two channels available for discussions and support. cFBA only needs the installation of CBMPy [75]. SteadyCom and MMT are already available as part of the COBRA Toolbox, whose installation is fast and easy. MICOM is easy to install as it only needs a sentence of code and the installation of COBRApy [76]. OptCom was run using the 'OptCom' function from MICOM, and thus, they were equally evaluated.

OptCom requires GAMS and BARON as solvers also accessed through GAMS. GAMS is not a free programming language, and therefore, its use is more limited. All the dependencies of cFBA and MICOM are freely accessible. SteadyCom, RedCom, MMT, and NECom are written in Matlab, which is not free, but licenses are inexpensive for a wide group of users. Besides, SteadyCom and MMT require COBRA Toolbox, and RedCom requires *CellnetAnalyzer* [77] that are both freely available. Required solvers are either integrated by default or freely accessible to a wide group of users. CASINO uses the RAVEN Toolbox, a free software suited for Matlab [78].

The developers of OptCom provide a book with tutorials to run the tool. Some tutorials are included in COBRA Toolbox that explains case studies used with SteadyCom. However, they merely explain some aspects of the software. MMT has explanatory tutorials and includes README files for each main function of the tool, but it is not completely maintained to have the latest functionality. MICOM contains an extensive user manual explaining all aspects of the tool, and it is up to date (at the time of preparing this manuscript).

All the tools require certain expertise from the user to be run. OptCom is not considered a user-friendly tool as GAMS is used as the programming language, and it is not an extended programming language. cFBA, MMT, and MICOM require some knowledge of programming and constrained-based modeling by the user. SteadyCom is somehow constrained to the case studies of the original publication, and thus, extending their use to new cases, and GEMs requires more knowledge by the user.

Except for CASINO and RedCom where the source code is unavailable, the source code is accessible to all users. OptCom, MICOM, and MMT produce the same results over time. OptCom, cFBA, and MICOM produce consistent solutions without infeasibility or non-convergence issues. SteadyCom and MMT can lead, in some cases, to non-convergence solutions or infeasibilities.

NECom was applied to model a co-culture of two species. OptCom and cFBA have been used to model small microbial consortia (up to 3). CASINO, MMT, and MICOM can be used to model large communities (gut microbiota). SteadyCom and RedCom can be applied to model larger microbial communities than OptCom and cFBA ($\approx 9$).

OptCom toolbox used 'txt' files as input models that do not meet any community-level standard. cFBA, CASINO, SteadyCom, RedCom, MICOM, and NECom meet SBML or COBRA community-level standards for model input and output files. Based on the original publication, we hypothesize that CASINO meets the SBML standard for model input and output since the RAVEN Toolbox is employed [78]. MMT meets both SBML and COBRA community-level standards for model inputs and outputs models in COBRA format. RedCom meets COBRA community-level standards. OptCom cannot be adjusted to standard GEM formats. cFBA uses CBMPy, which reads the standard SBML format. MICOM is used with COBRApy, which allows for different GEM formats. CASINO reads models imported in COBRA format. SteadyCom allows for the translation of GEM models from/to other formats. MMT allows for the import of SBML, XML, and COBRA models within the tool, and the user does not need to transform them before running the tool. NECom allows for the use of COBRA models.

Every tool/approach allows for adapting some parameters and constraints with more or less difficulty. MMT and MICOM provide a wide set of visualizations for different analyses. The rest of the tools do not produce visual outputs by default.

## Quantitative assessment—Static tools/approaches

cFBA, SteadyCom, MICOM, and MMT scored higher than the other tools in the qualitative assessment (see S1 Fig), and thus, they were also evaluated regarding their performance in reproducing an experimental case study. OptCom was also assessed because it was available as an additional function in MICOM, and it is highly cited (S1 Table). The case study consisted of syngas fermentation to medium-chain fatty acids by a co-culture of *C. autoethanogenum* and *C. kluyveri*. Syngas is a mixture of $H_2$, CO and $CO_2$. *C. autoethanogenum* can thrive on syngas by assimilating CO or $CO_2$ and $H_2$ and producing acetate and ethanol as byproducts. *C. kluyveri* does not grow on syngas. It needs acetate and ethanol to grow, and therefore, it depends on the direct cross-feeding of these metabolites by *C. autoethanogenum*. The study of Diender et al. [9] reported data on the steady-state consumption and production rates of CO, acetate, ethanol, butyrate, and caproate. The hydraulic retention time (HRT) of the reactor was fixed in the chemostat, suggesting that the species growth rates and the community growth rate were equal (Fig 3A).

The latter assumption is also considered in cFBA, SteadyCom, and MMT, as illustrated in Fig 3A. cFBA and MMT were run constraining the growth rates to the experimental values. SteadyCom allows for the definition of the species' growth rates and relative abundance. However, SteadyCom was not feasible in those conditions, and the unconstrained parameters resulted in higher growth rates than the experiments. OptCom and MICOM do not assume

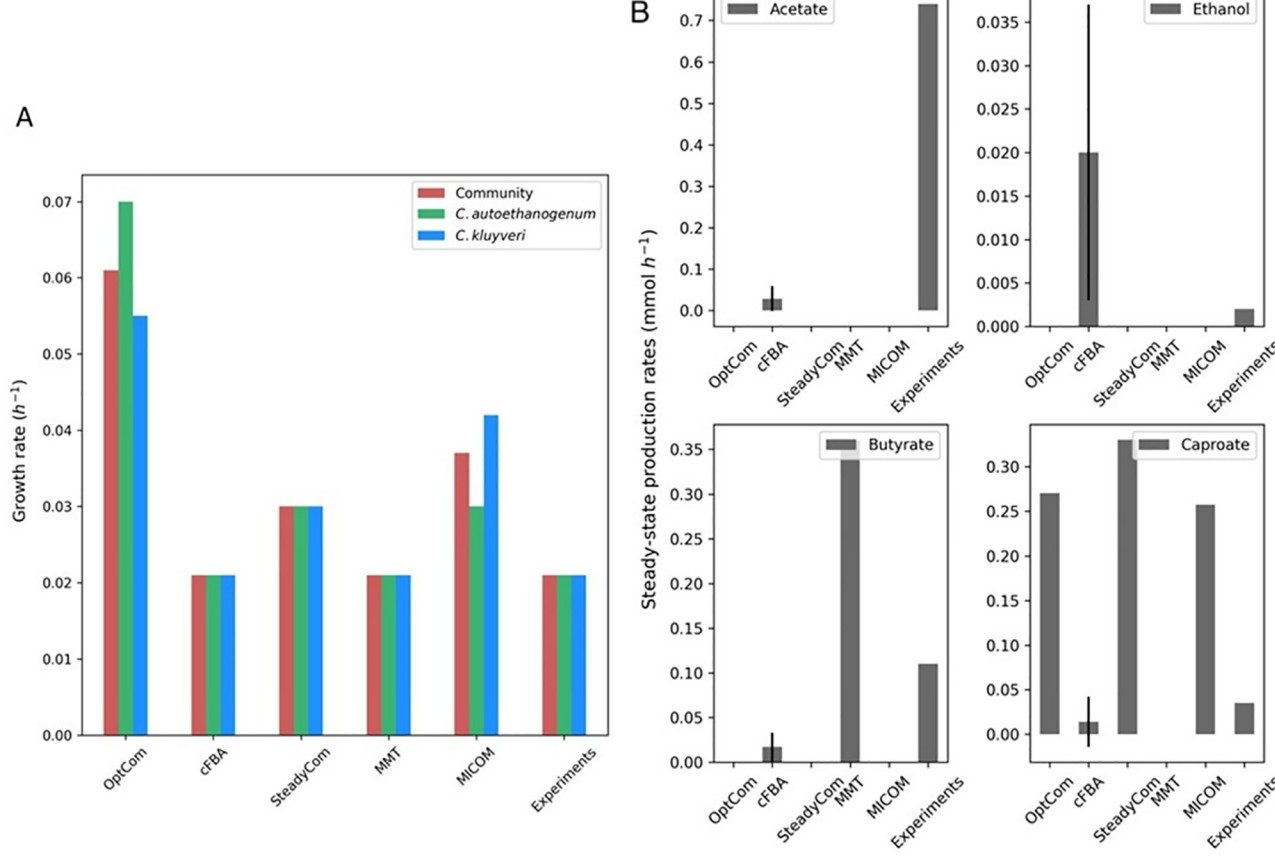

**Fig 3. Comparison of tool predictions to experimental data from the study of Diender et al. [9].** (A) Community and species growth rate and (B) Steady-state production rates of the main fermentation products obtained in the fermentation of CO by *C. autoethanogenum* and *C. kluyveri* with the assessed tools. cFBA shows the average fluxes and standard deviation of the samples used with Flux sampling.

equal growth rates of species, and the predicted growth rates were also higher than the experiments. The use of additional constraints in OptCom and MICOM to enforce equal growth rates is possible and would result in a better fit, but tools have been run using the standard methodology.

In the study of Diender et al., almost all the ethanol produced by *C. autoethanogenum* on CO was fully consumed by *C. kluyveri*, whereas the acetate produced by *C. autoethanogenum* was only partly consumed by *C. kluyveri* (Fig 3B) [79]. The fermentation of acetate and ethanol by *C. kluyveri* led to butyrate ($\approx 0.11$ mmol h$^{-1}$), caproate ($\approx 0.035$ h$^{-1}$), and H$_2$ (not showing here). Standard deviations for experimental values were not reported.

All tools predicted the exchange of acetate and ethanol between *C. autoethanogenum* and *C. kluyveri*, except MMT, which only predicted the exchange of ethanol (see fluxes in the online repository: https://doi.org/10.5281/zenodo.8074832). OptCom, SteadyCom, and MICOM predicted that the acetate produced by *C. autoethanogenum* was entirely consumed by *C. kluyveri* (see fluxes in the online repository: https://doi.org/10.5281/zenodo.8074832. However, cFBA predicted acetate production by the co-culture (Fig 3B) and acetate consumption by *C. kluyveri*. Yet, acetate production was lower compared to the acetate measured in chemostat ($\approx 0.73$ mmol h$^{-1}$). MMT did not predict the production of acetate. Ethanol was difficult to detect in chemostat ($\prec 0.1$ mM), as most of it was consumed by *C. kluyveri*. cFBA predicted more ethanol than in chemostat, and in OptCom, SteadyCom, MMT, and MICOM, ethanol was fully consumed by *C. kluyveri*. Butyrate production was only predicted by cFBA and MMT. Caproate production was over-estimated by all the tools except cFBA, which predicted a value comparable to the experiments, and MMT, which did not predict caproate production.

Following the procedure of the modeling study of the same co-culture [79], flux sampling was used to compute the fluxes with cFBA. While the agreement of experimental data with model predictions is less accurate here than in the previous computational study for the lack of constraints, the new adaptation of cFBA can reproduce the experimental data better than the other tools [9]. MICOM, SteadyCom, and OptCom could also reproduce the cross-feeding of acetate and ethanol, which is a key feature in this co-culture. MMT only reproduced the exchange of ethanol, being infeasible for the CO uptake rate reported in the experiments. The adjustment of the coupling factor ('c') or the use of mgPipe pipeline [46] in MMT could lead to alternate outputs. Regardless, MMT was mainly designed to model the gut microbiome using AGORA models, and thus, it is more suitable to model large communities. The use of FVA as an alternative method to compute fluxes with the existing tools and approaches could also lead to additional information on cross-feeding metabolites.

## Qualitative assessment—Dynamic tools

Fig 4 shows the evaluation of the described features in every tool based on the evaluation rubric (S2 Table). From the qualitative assessment of all the dynamic tools, it was apparent that none were flawless (all features being good or excellent). There were, in fact, compromising aspects in many of the tools where tools were superior for some features while flawed for other features (see Fig 4 and S2 Fig). For instance, when examining their availability, it was determined that dyMMM, DAPHNE, MMODES, and surfinFBA were good or excellent. Still, the tools were considered merely satisfactory when evaluating their interoperability, flexibility, and reproducibility, except MMODES. In addition, tools such as dyMMM, DFBAlab, μbialSim, and MMODES were easy to install and contained accessible and tractable dependencies, but did not score above satisfactory levels for the software being traceable (Fig 4). Nevertheless, there was one tool, d-OptCom, that scored poorly compared with the other methods because

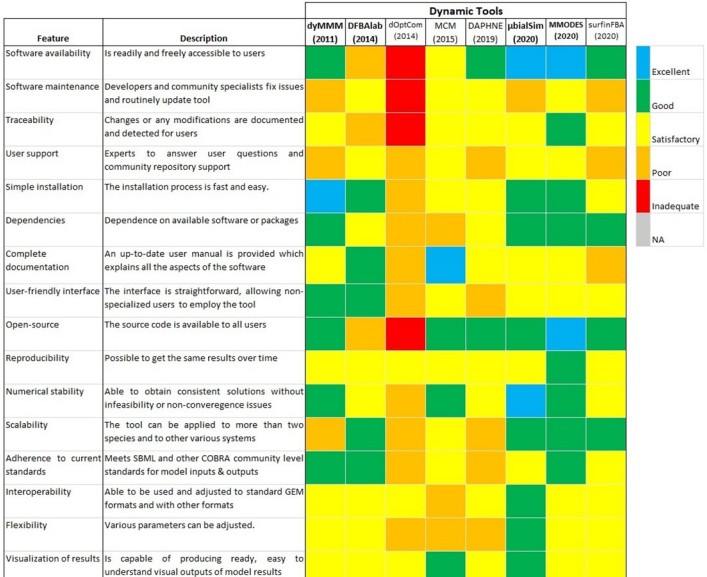

**Fig 4. Qualitative assessment of the dynamic tools.** Colored squares indicate the evaluation of the specified feature in every tool. The color scale (upper right) goes from excellent (blue) to inadequate (red). When a feature does not apply to the specified tool or the feature was not evaluated, it is indicated as NA (Not applicable; grey). The metrics contained in the figure were inspired by [47]. The tools are ordered by year of publication.

the software was not readily available from the developers (available upon reasonable request). It was also constructed under an uncommonly utilized GAMS programming language.

Most of the tools received a good or excellent score for some features. For example, all the tools, except d-OptCom and DFBAlab, are open source (contain source code available to all users). DFBAlab also includes source code reachable to some users, but potential users first need to request the software to gain access as it is not available in the public domain for all users. All tools were deemed at least satisfactory regarding their reproducibility and potential to visualize outputs. Furthermore, all the tools other than d-OptCom and DAPHNE adhere to COBRA community standards [80] and contain a user-friendly interface, at least to a satisfactory degree. This means most tools consist of relatively simple and direct ways of designating inputs, such as adding GEMs, setting up media compositions, and defining model constraints and kinetic parameters. Also, tools that were built incorporating existing COBRA frameworks, such as COBRApy or COBRA Toolbox, typically scored well for being user-friendly and following current standards because they assimilate the current community standards and acquire greater functionality from the existing infrastructure.

All the dynamic tools received at least a satisfactory score for their interoperability except MCM, which got a poor score. This is due to MCM having difficulties handling some GEMs, especially those of eukaryotes with multiple compartments. Also, MCM utilizes an in-house convention when processing GEMs; thus, issues could arise when needing to employ the GEM in another tool or database. It is best that GEMs adhere to SBML standards [81]. Most software except MCM and d-OptCom earned at least a satisfactory score for dependencies. MCM earned a relatively low score because of the reliance on a not up-to-date micog.py extension (Python 2.7) for using GEMs. Overall, tools such as MMODES, μbialSim, and dyMMM received superior scores because they are readily available, accessible, and inoperable compared to the other tools.

### Quantitative assessment—Dynamic tools

A case study comprising a dataset published by Hanly and Henson [73] was used to validate and quantitatively evaluate the dynamic tools. The experimental setup consists of a bioreactor operating in batch mode with a co-culture of *E. coli* and *S. cerevisiae* designed to study the efficient aerobic consumption of glucose/xylose mixtures. Each microbe utilizes a specific substrate. *S. cerevisiae* only consumes glucose, whereas the engineered *E. coli* strain ZSC113 only consumes xylose. This was done to prevent diauxic growth shown in monoculture for *S. cerevisiae*. The individual biomasses of the two microbial species as well as the concentration dynamics of the substrates glucose and xylose, were studied. Moreover, ethanol concentration dynamics were also measured because *S. cerevisiae* produces it during fermentation, thus, inhibiting the growth of *S. cerevisiae* and *E. coli*.

Furthermore, tools that scored 50 or above when summing up the qualitative scores were deemed of sufficient quality for a further quantitative examination. These tools were DFBAlab, dyMMM, µbialSim, and MMODES. We evaluated each tool's capability of simulating the observed kinetics using the same inputs across the tools. The error distribution for the different methods was determined (see Methods Section for a description of the normalized error calculation). *S. cerevisiae* and *E. coli* biomass concentration increases (Fig 5). DFBAlab was the only tool to predict the biomass formation of *E. coli* somewhat accurately though the simulation differed substantially from hours 5 to 12.5, showing no growth while growth was observed (Fig 5B). µbialSim and MMODES predicted growth for *E. coli*, but the biomass formations

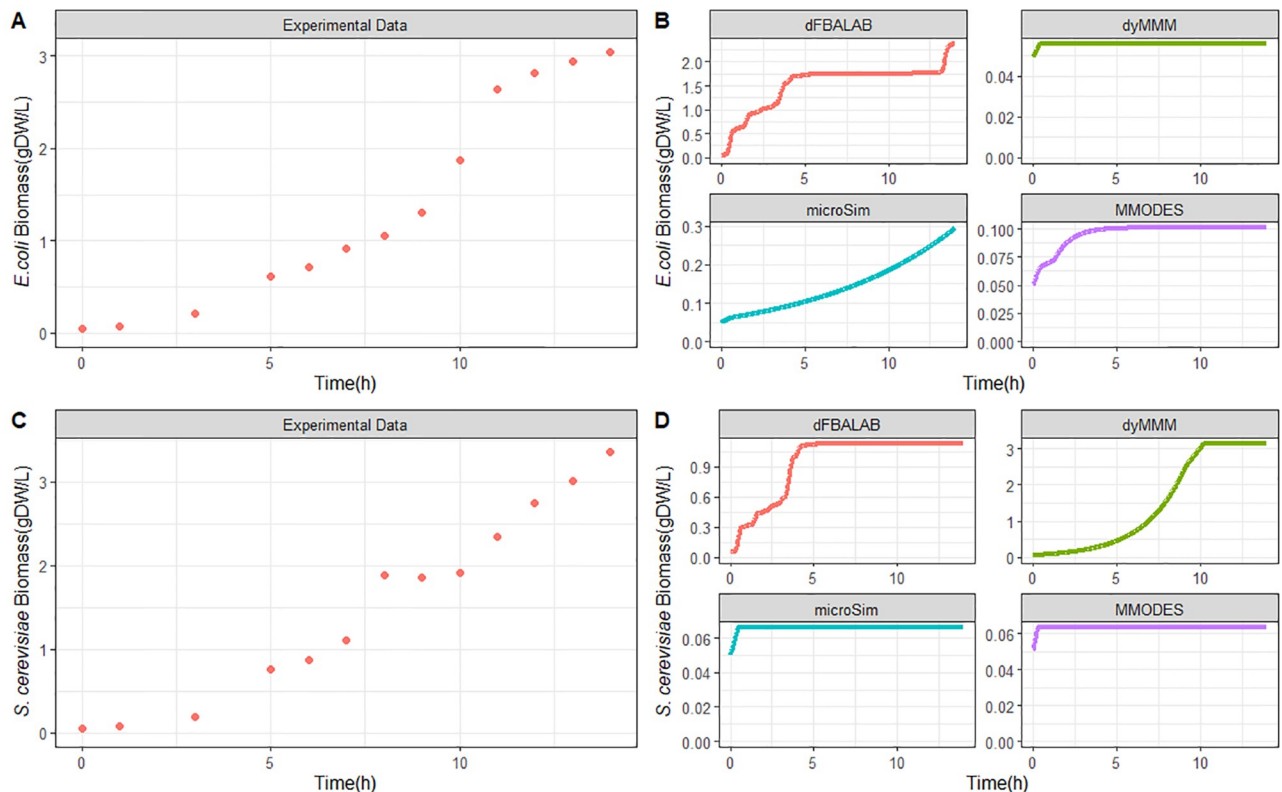

**Fig 5. Quantitative assessment of the dynamic tools.** Experimental biomass concentration profiles of *E. coli* and *S. cerevisiae* (Panel A and C, respectively). Comparison of tool predictions of biomass concentration profiles with data for *E. coli* and *S. cerevisiae* (Panel B and D, respectively) from the study of Hanly and Henson. [73].

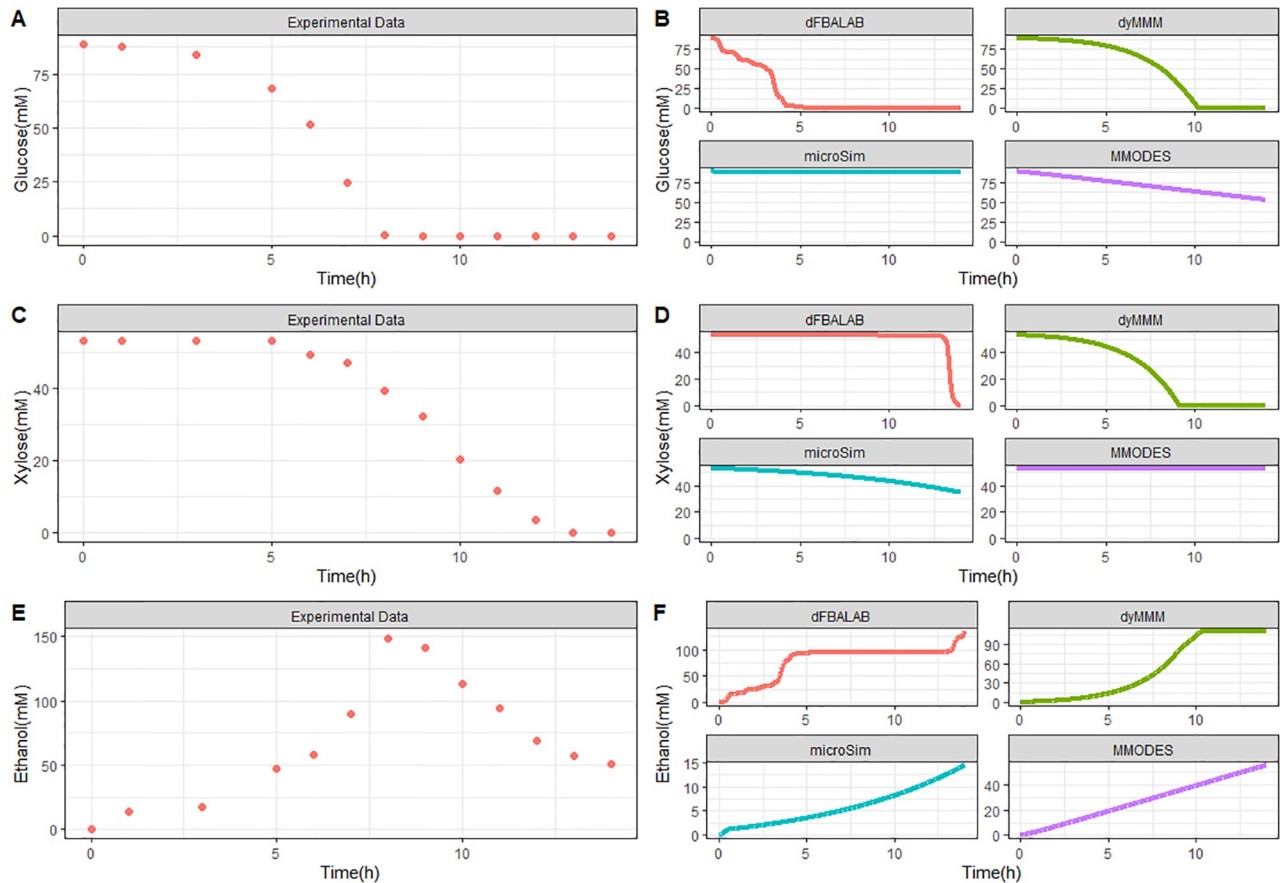

**Fig 6. Quantitative assessment of the dynamic tools.** Experimental extracellular metabolite concentration profiles of glucose, xylose, and ethanol (Panel A, C, and E, respectively). Comparison of tool predictions of extracellular metabolite concentration profiles with data for glucose, xylose, and ethanol (Panel B, D, and F, respectively) from the study of Hanly and Henson. [73].

were at least an order of magnitude below measured biomass levels (Fig 5B). dyMMM predicted slight growth of *E. coli*. Overall, µbialSim and DFBAlab achieved the best predictions for *E. coli* based on the experimental data with $R^2$ values of 0.959 and 0.775, respectively. For the *S. cerevisiae* growth kinetics, the dyMMM tool most accurately simulated biomass formation (Fig 5D). While DFBAlab predicted growth for *S. cerevisiae*, the final biomass concentration was about 65% less than the experimental value (Fig 5D). µbialSim and MMODES simulated slight growth for *S. cerevisiae*. Overall, dyMMM and DFBAlab achieved the best predictions for *S. cerevisiae* based on the experimental data with $R^2$ values of 0.900 and 0.682, respectively (for all $R^2$ values, see S1 Data).

The measure and predicted kinetics for the consumption of glucose and xylose, as well as the formation of ethanol, are illustrated in Fig 6. Both sugars' concentration decrease over time, where glucose is completely consumed around 7.6 h while xylose remains in the medium longer and is used up entirely by around 13.2 h. This is due to a faster glucose uptake rate from *S. cerevisiae* compared to xylose's consumption rate by *E. coli*. There is a steady production of ethanol until the 7.5 h duration. Then, there is a decrease in ethanol, perhaps because *S. cerevisiae* consumes ethanol under aerobic conditions. The co-culture featured a high increase in ethanol levels due to *S. cerevisiae* fermentation metabolism; ethanol is an inhibitor of *E. coli*

growth. This ethanol inhibition from *S. cerevisiae* on *E. coli* growth was the only interspecies interaction identified in this experiment.

For the sugar substrates glucose and xylose, dyMMM and DFBAlab simulated the kinetics most precisely according to the experimental values (Fig 6A and 6B). However, DFBAlab predicted rapid xylose consumption at the end of fermentation. MMODES simulated a linear glucose consumption with a final concentration of about 52 mM, while xylose consumption was not predicted. μbialSim did not simulate glucose consumption, while xylose was consumed slightly with a final concentration of about 35 mM. All the tools simulated ethanol production. Nevertheless, none of the tools could accurately estimate the dynamics of ethanol production. This is because the diauxic shift from glucose to ethanol by *S. cerevisiae* cannot be modeled by simple optimization as done by FBA. dFBAlab predicted the magnitude and kinetic characteristics most accurately, while μbialSim and MMODES gave an underestimation for the ethanol concentrations and reflected linear profiles (Fig 6C). Overall, DFBAlab achieved the best predictions for glucose, xylose, and ethanol kinetics based on the experimental data with $R^2$ values of 0.902, 0.978, and 0.440, respectively (for all $R^2$ values, see S1 Data).

## Qualitative assessment—Spatiotemporal tools

Fig 7 shows the evaluation of the described features in every tool based on the evaluation rubric (S2 Table). From the qualitative assessment of all the spatiotemporal tools, none of the tools received a perfect score. Many tools were highly rated in many categories, while several were also deficient in many categories (see Fig 7) and S3 Fig). For example, COMETS, BacArena, CROMICS received excellent scores for software availability and being open-source because these tools and their source codes are readily available via a GitHub repository or a maintained website, whereas MatNet received an inadequate score because neither the tool nor its source code can be easily accessed online. When evaluating the software maintenance,

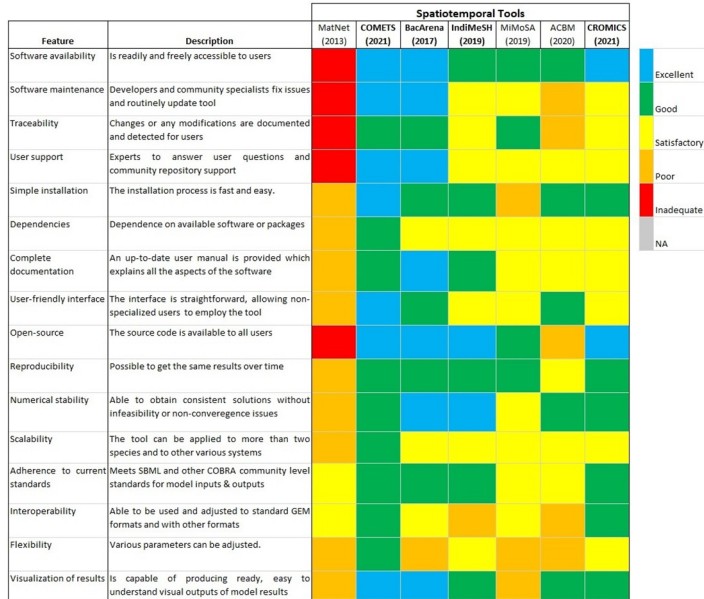

**Fig 7. Qualitative assessment of the spatiotemporal tools.** Colored squares indicate the evaluation of the specified feature in every tool. The color scale (upper right) goes from excellent (blue) to inadequate (red). When a feature does not apply to the specified tool or the feature was not evaluated, it is indicated as NA (Not applicable; grey). The metrics contained in the figure were inspired by [47]. The tools are ordered by year of publication.

user support, and ease of installation of the software, two tools stood out as excellent: COMETS and BacArena. In contrast, many of the tools, including IndiMeSH, MiMoSA, ACBM, and CROMICS were merely rated as being satisfactory to poor for software maintenance and user support (Fig 7). However, except for MiMoSA, they were regarded as good for simple installation because of the accompanying instructions within the software download. Most of the tools received a good or excellent score for some features. For instance, for reproducibility, scalability, and visualization of results, all tools scored good to excellent, except ACBM and MatNet (for reproducibility), MiMoSA and MatNet (for scalability), and MiMoSA and MatNet (for visualization of results) (Fig 7). ACBM did score well for reproducibility because slight changes in the same GEM files caused the tool to be invalid or inoperable. In general, IBM methods scored lower in terms of scalability because of the additional computational costs compared to PLM methods. Most of the tools contained some functionality for visualizing results or rendered outputs which were easy to use by other software tools. Many of the tools have at least a satisfactory level of traceability, reliance on available software, and documentation. Only ACBM scored poorly for traceability because changes to the software cannot be detected, whereas most other tools contain mechanisms to trace various versions, and updates are documented.

All the spatiotemporal tools received at least satisfactory scores except MatNet for their user-friendly interfaces and numerical stability (Fig 7). These relative scores reflect how well the functionality, design, and incorporation of community-devised constraint-based modeling infrastructure into the tools. Also, some tools, such as COMETS and ACBM contain GUI options for novice users who may be less comfortable using command-line interfaces. Many of the tools were numerically stable, using examples implying users can start using the software assuming no infeasibilities occur under basic conditions. Overall, tools such as COMETS, BacArena, and CROMICS received superior scores because they are more readily available, up-to-date, open-source, and contain user-friendlier interfaces compared to the other tools.

## Quantitative assessment—Spatiotemporal tools

A case study comprising a dataset published by Harcombe et al. [33] was used to validate and quantitatively evaluate the spatiotemporal tools. The experimental setup consists of a two-member consortium of two mutant strains of *E. coli* K-12 and *S. enterica* LT2, designed to study the syntrophy between the two strains. *E. coli* K-12 is deficient in methionine production, so it relies on methionine production from *S. enterica* LT2. Conversely, *S. enterica* LT2 relies on the secretion of acetate by *E. coli* K-12 because *S. enterica* LT2 cannot uptake lactose under microaerobic conditions. This experiment created the optimal condition to observe a mutualistic relationship where neither species can grow without the other being present. The two-species consortium was grown as a lawn on a Petri dish. Furthermore, experimentally, *E. coli* and *S. enterica* were grown overnight in permissive lactose Hypho minimal media (see S2 Table in [33] for more details) and then mixed at a ratio of 1:99 and 99:1. To examine the impact of time and space on the consortium, on LB, both *E. coli* and *S. enterica* can grow independently, and X-gal(5-bromo-4-chloro-3-indolyl-b-D-galactopyra- noside) was included in the plates so that blue *E. coli* colonies could be distinguished from white *S. enterica* colonies (see [33] for more details). The individual biomasses of the two microbial species were examined and measured over a 48-hour growth cycle by counting colonies. By the end of the cycle, the composition converged even when the inoculum frequencies varied by two orders of magnitude. Furthermore, the study also found a relationship concerning the spatial structure with the metabolite resources being allocated, which caused decreased growth between the species as they were moved further apart.

From the qualitative assessment, there were four tools considered to be adequate for a further quantitative survey. Furthermore, from an enumeration of the qualitative scores, tools that scored overall 50 or above were deemed requisite quality (see S3 Fig). These tools were COMETS, BacArena, IndiMeSH, and CROMICS. We evaluated each tool's capability of simulating the observed biomass dynamic as well as the spatial composition based on moving the species apart using similar inputs across the methods. The error distribution for the different methods was determined (see Methods Section for a description of the normalized error calculation). Over the 48-hour cycle, the community composition of species converged regardless of whether the initial frequency was 1% or 99% of *E. coli*. COMETS predicted species ratios for the two initial frequency conditions (Fig 8). COMETS predicted a composition of 79% ± 4% *E. coli*, which is not significantly different from the experimental frequency of 78% ± 6% (Fig 9A). However, BacArena predicted a composition of 63.2% *E. coli* (Fig 8B). The predictions of the community composition from IndiMeSH were determined not to be significantly different from both the experimental and COMETS simulation results (p-value of 0.66 and 0.33, respectively, two-tailed t-test). The CROMICS results showed that regardless of the initial frequency, the system converged in a species ratio of 76.3% ± 0.1% for *E. coli*.

The two-species consortium was used to examine the influence of spatial structure on resource allocation and growth within this mutualistic system. Here, we illustrate the evolution of the colony spatial distribution over time using a PBM (COMETS) and an IBM (BacArena) for their representations of biomass (Fig 9). The COMETS simulation shows a rapid rise in the population density of *E. coli* and a shift in the location within the grid from *E. coli* from 9h to 48h (Fig 9A). COMETS also predicted a moderate increase in population density in *S. enterica* as well as a slight shift in position from 9h to 48h (Fig 9A). BacArena takes a different approach and models each organism individually on a two-dimensional grid to simulate a spatial environment. For BacArena, the colony evolved first around the initial microbial positions of *E. coli* and *S. enterica* where *E. coli* reaches a near-final individual amount t = 9h (Fig 9B). The simulated *S. enterica* colony continued to grow more abruptly, however, from t = 9h to t = 48h, filling up most of the remaining spatial elements (Fig 9B).

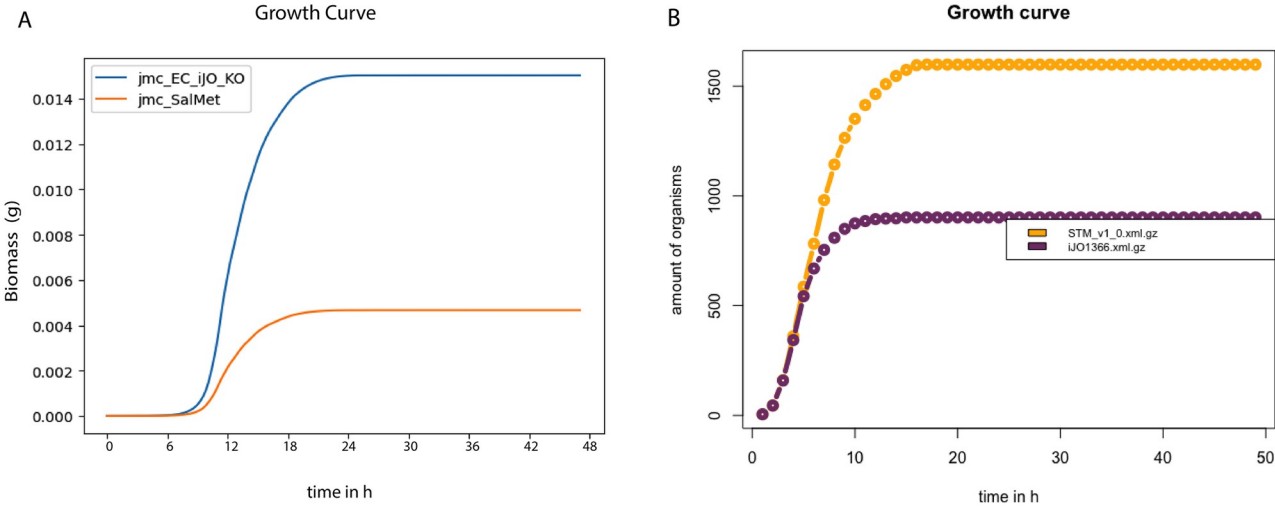

**Fig 8. Quantitative assessment of the spatiotemporal tools.** Abundance simulation profiles of *E. coli* and *S. enterica* from COMETS (Panel A) and BacArena (Panel B). Note biomasses used per individual organisms were $5 \times 10^9$ fg and $3 \times 10^8$ fg for *E. coli* and *S. enterica*, respectively. Experimental data used was from the study of Harcombe et al. [33].

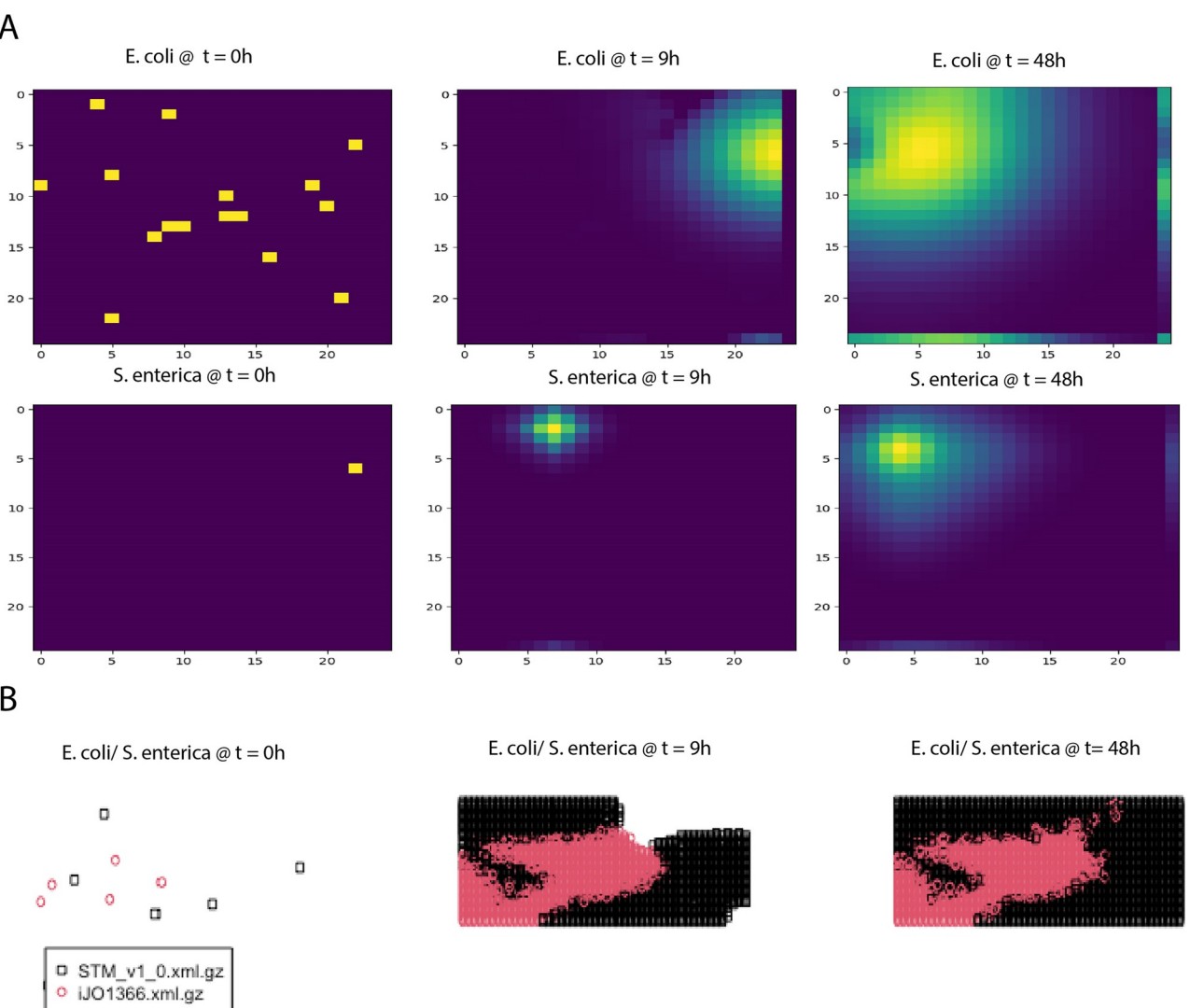

**Fig 9. Quantitative assessment of the spatiotemporal tools.** Colony spatial distribution over time simulation profiles of *E. coli* and *S. enterica* from COMETS (Panel A) and BacArena (Panel B). For Panel B, the red dots represent *E. coli* and the black dots represent *S. enterica*. Experimental data used was from the study of Harcombe et al. [33].

As a simple test, we attempted to simulate conditions where the growth of the species should be affected by increasing distance between them. Consistent with what was observed experimentally [33], the simulations using COMETS, IndiMeSH, and CROMICS showed similar degrees of decreased growth as they were initiated further apart (see Zenodo repository: https://doi.org/10.5281/zenodo.8074832 for output data for all the tools). However, the BacArena tool could not capture any variation in spatiotemporal growth as observed during the experiment.

## Discussion

This study provides a survey of the chronicled expansion of GEM modeling tools/approaches for microbial communities throughout the past decade and recommends the best GEM modeling tools based on qualitative and quantitative assessments. In particular, we assessed tools

that used CBM methods and their performance to model synthetic microbial communities of two species. This study also illustrates the challenges and brings forth issues and recommendations that should be considered when developing future tools, so that overall serviceability and performance can best facilitate potential users.

In general, GEM modeling tools fall into several categories for analyzing microbial communities. They have been used to identify interspecific interactions, which are embodied in tools such as SMETANA [82]. These tools can also contain extensive functionality, as illustrated in the Community Gap-Filling [83] tool for creating draft GEMs from genome assemblies and performing gap filling and, based on the resolved metabolic gaps, can determine interspecies relationships. There are tools available such as CommModelPy [84], DOLMN [20], FLYCOP [85], and Community Opt-yield-FBA [86], which were developed to facilitate the synthetic designing of microbial consortia. However, this study focuses on static, dynamic, and spatio-temporal Flux-Balance-Analysis-based modeling tools used for simulating extracellular and intracellular metabolic phenotypes of microbial consortia. Furthermore, the scope of the work was limited to evaluating GEM modeling tools for microbial communities where the user needs to provide draft or curated GEMs as inputs and not tools that are used primarily to create the reconstructions. CommModelPy [84], DOLMN [20], and FLYCOP [85] likewise were omitted from this study's assessment because we believe they are outside its scope. For instance, the DOLMN tool is applicable for investigating the division of labor or interactions among different strains within the same species. Moreover, the FLYCOP pipeline employs the COMETS version 1 tool, which is already included in this study's assessment. Therefore, we feel that FLYCOP is more appropriate as an expansion tool to facilitate the *in-silico* design of synthetic microbial communities.

The criteria to qualitatively evaluate the tools was based on the FAIR principles that we consider, should be applied for data, operating procedures, tools, and models [48, 50]. Generally, tools that scored the highest in the qualitative assessment were shown to perform best in the quantitative assessment, highlighting the importance of following the FAIR principles. Tools that were less 'FAIR' could hardly be accessed nor used in this study. Thus, further information and access to the code would be required to quantitatively assess the performance of these tools and draw a conclusion. Nevertheless, a tool needs to be FAIR to be used, to contribute to the field, and thus, to shed light on the continuous challenges that can be, later on, tackled in new tools. In fact, this study observed the latter, since the continual publication of GEM modeling tools for microbial consortia over the past decade has generally improved in performance over time. The first tools have served as a proof of concept and have established the basis for developing the new tools. However, there were a few exceptions of tools that had high scores in the qualitative assessment and performed worse than the average in the quantitative assessment.

In this study, the assessment was based on the modeling of two-species co-cultures, and therefore, the results could have varied when the tools were applied to larger communities, or to alternative systems. In addition, the limited availability of experimental data affected the selection of case studies, which might have an effect on the assessment of tools. By having more publicly available datasets, GEM modeling tools would be able to be more robustly evaluated on a quantitative level in terms of their scalability and numerical stability. In general, the performance of these tools would be enhanced if the quality of the GEMs or model architecture was improved, and more constraints were applied to fit the experimental studies or to compute fluxes. However, we ran the tools without augmenting them and following the reported standards used to avoid an unbiased evaluation. The quantitative evaluations should be taken with a degree of awareness about their generality and projection to other types of systems. Furthermore, the quantitative evaluations here performed should be carefully considered

as indications of tool performance rather than absolute measures of performance. They represent the behavior of the tools when confronted with specific datasets with defined biological characteristics. The syntrophic relationship established in the system used to test the static tools is different from the alleviation of toxicity observed in the dataset used for the dynamic tools. We cannot rule out that tool performance is affected by the nature of the interactions.

The current status of the array of static, dynamic, and spatiotemporal tools for GEM modeling microbial consortia presents a substantial barrier to entry for researchers unfamiliar with specific requirements and idiosyncrasies that may arise within CBM frameworks. Tools should be user-friendly enough for a novice user who may be familiar with metabolic modeling concepts such as FBA and has some basic programming knowledge. We believe that user-friendliness is also linked to the quality of available documentation and the existence of community support. Therefore, we recommend keeping manuals and related repositories up to date and including dependencies amenable to changing computational environments or, alternatively, describing which packages are compatible. When the software is open/free, a possible solution could be to provide a container with the tool and the dependencies. All changes should be equally specified and updated in all places where the tool can be found unless specified otherwise. It is crucial that the tool is accessible to the users. Licenses are also important and should be provided because they allow tools to be incorporated in pipelines such as FLYCOP that build upon and extend the functionalities of COMETS to design microbial communities. We recommend the tools can read several GEM formats and namespaces without the need to translate the input models using other packages (e.g., COBRA Toolbox, COBRApy). The translation of these models often leads to a loss of information and a lack of key attributes that might generate errors when running the tools. The community models built by some static tools could keep the original bounds of the transport reactions of a single GEM to avoid the user the re-definition of the bounds once the community model is built. This would extend the use of these tools to cases with unspecified mediums/diets. Dynamic tools should allow the possibility of including inhibitory effects to better predict community behavior.

Tools oriented for modeling small and big communities should incorporate sufficient series of examples featuring a range of application scenarios from small to big communities. Tools could incorporate case studies with GEMs of non-model organisms since many case studies are typically proven on communities of several *E. coli* species that use the same GEM. Other tools are meant to work with models retrieved from a specific database and defined namespace (e.g., AGORA [87]). The applicability of these tools to non-model organisms is critical for designing and optimizing synthetic communities. In the past decades, many synthetic communities have been established that do not depend on model organisms and have shown a high potential for producing commodity chemicals [4, 88–91]. On the other hand, GEMs require accurate genome annotation and curation to properly account for interactions mediated through pathways, but there is still difficulty identifying these components in microbial genomes. Therefore, most GEM modeling efforts have been restricted in application to high-quality GEMs (*E. coli* and *S. cerevisiae*). To extend the applicability to alternative GEMs, the quality of GEMs of single species and communities should follow the community standards (e. g standard-GEM [92]) and, thus, be verified using test suites such as MEMOTE [93].

Some tools are based on approaches that use FBA and, therefore, are limited to the maximization of the community growth rate or species growth rate. In those cases, FBA favors growth over production, and cross-feeding metabolites might be overlooked (Fig 3B). Objective functions can be improved by using experimental data, or frameworks can be refined in dynamic tools to use a bi-level objective routine [94] or a multiphase, multiobjective approach [95]. Yet, these amendments may still not apply to all growth scenarios [96]. Flux sampling has been proven to be an effective tool to explore metabolism using GEM [31, 97, 98], and its use can

mitigate the influence of the objective function. Flux sampling has also been successfully implemented to model microbial communities [79]. Therefore, we recommend integrating flux sampling methods in current and newly developed tools for modeling microbial communities in different environments, even though computational costs are high [99].

Future tools/approaches could consider using enzyme-constrained models to model microbial communities since they have shown great potential to better understand microbial phenotypes and phenomena such as overflow metabolism [31].

It is of the utmost importance that static tools/approaches incorporate the biomass composition of each species in their formulations beyond simply integrating the ratio by changing the stoichiometry of their biomass reaction. There is no clear consensus yet among static approaches on whether the species grow at balanced growth or whether the community and species growth rates differ. This challenges the selection of the objective function and the optimization approach of static tools. We believe that static tools best fit the simulation of continuous environments where the dilution rate is assumed to be constant, and thus, the community growth rate should be the same as single species [79]. Techniques that predict variable growth rates among microbes within a community should be incorporated into dynamic tools. The drawback, nonetheless, is the inability to account for longitudinal community composition. An original method for alleviating this limitation is integrating ecological models within existing constraint-based frameworks. Some researchers have already begun employing COBRA methods and evolutionary game theory [100]. However, in the future, it would be beneficial to bolster more tools in this direction.

The results of this work suggest there is still much room for improving the overall quality of GEM modeling tools for microbial communities. Despite the immense number of proposed tools over the last decade, the availability, user-friendliness, and overall performance quality of the tools are disparate, and there is no perfect tool for all scenarios. In addition, although some of the tools were available and accessible to a moderately novice-level user, there were still some flaws in the quantitative predictions of those tools. However, we determined some tools were exceptional and should be preferred starting points for researchers or developers.

For instance, for static systems, cFBA is the approach that achieves the best outcome, but it largely requires manual adaptation. Moreover, it is not a tool, but merely a generalized approach or algorithm. However, self-contained tools, such as MICOM and SteadyCom, produce reasonable results. None of them outperform the others, so they can be considered as a starting point, and given the usability characteristics, we recommend MICOM. For dynamic systems, the choice is more difficult to discern, but given our results, we suggest DFBAlab. MMODES is also a solid option, although discretion is advised when interpreting results from MMODES as predictions largely disagree with the measurements. Yet ethanol inhibition should be taken into account because the case we simulated contains ethanol inhibition. Therefore, an option or flexibility for augmenting a more complex Michaelis–Menten or Hill kinetic expression to reflect growth rate suppression at high ethanol concentrations would be desirable in a new tool. For spatiotemporal systems, we recommend either COMETS or BacArena because of their extensive development, accessible platforms, and documentation. COMETS performed better in simulating a simple co-culture experiment. However, BacArena is an individual-based method that can be beneficial when studying heterogeneous cell populations.

Through this qualitative and quantitative survey, we have presented and analyzed a broad overview of many GEM modeling tools for microbial consortia. These tools have been successfully applied in various types of systems of microbial communities. In addition, while these constraint-based tools have significantly transformed our understanding of communities by incorporating mechanistic details and highlighting metabolic interactions, there are still many

opportunities for improving modeling frameworks by making software FAIR, user-friendly, and improving the accuracy of simulations of various types of systems. The work presented here can guide researchers in selecting the proper modeling tools and help developers build upon suitable modeling frameworks for new software tools.

## Materials and methods

An extensive literature review was executed to identify the state of the art of tools for modeling microbial communities in steady-state, dynamic, and spatiotemporal environments that used GEMs. Every tool was rigorously studied to follow a qualitative assessment. For that, we created an evaluation rubric that described each of the evaluation levels for every feature (S2 Table). All tools evaluated in this study are the most up-to-date versions as of December 15, 2022. In addition, we looked into studies that provided experimental data of synthetic microbial consortia of two species in every environment and assessed the performance of a set of each type of tool to reproduce them. The code used to produce our reported results and figures can be downloaded from: https://doi.org/10.5281/zenodo.8074832 and https://gitlab.com/wurssb/Modelling/modelingtools_microbial_consortia. The scripts can be used by other users as a guide to facilitate the evaluation and use of these tools.

### Evaluation of static tools

The case study chosen to evaluate the static tools was the production of medium-chain fatty acids from the fermentation of CO by a co-culture of *C. autoethanogenum* and *C. kluyveri*. The GEM of *C. autoethanogenum*, iCLAU786 [101], and the GEM of *C. kluyveri*, ickl708 [102], were downloaded from their original publications in SBML (xml) format and used as the input files in every tool. Extracellular metabolites (defined in compartment '_e') and exchange reactions ('EX_XX') common in both GEMs were modified to have the same namespace, as required to build the community model in all tools.

Experimental data was obtained from steady-state concentrations of fermentation products, total biomass, and hydraulic retention time (HRT) reported in the study of Diender et al. [9]. In particular, we simulated the condition defined in reactor run number three from the latter publication (Tables 1 and 3; 116 mmol CO L$^{-1}$ d$^{-1}$ and 0 H$_2$). CO feed rate and steady-state concentrations of fermentation products were converted to mmol h$^{-1}$, and HRT (d) was converted to growth rate (h$^{-1}$). The biomass species ratios were obtained from the computational study of the same co-culture [79], 0.4–0.6 *C. autoethanogenum-C. kluyveri*, respectively. CO feed rate was constrained to the experimental value in every tool unless stated otherwise. Additional constraints were applied and are specified per tool in the S3 and S4 Tables. Predicted steady-state fluxes of acetate, ethanol, butyrate, and caproate, as well as the community and species growth rate, were compared to the experimental measurements.

Below, we describe the specific methods applied to run every tool. The links to access the tools repository can be found in S1, S5 and S9 Tables.

**OptCom.** OptCom was run using the OptCom function found within the MICOM module called 'micom.optcom' instead of using the code from the original publication. Therefore, we first installed MICOM 0.32.2 using pip upon installation of Python 3.7 and COBRApy 0.24.0. 'OSQP' was installed by default with MICOM and used as solver. The community model was created using the 'Community' function. The function requires a 'taxonomy' table as an input parameter, which contains information about the species, models, and species relative abundance (biomass fraction). In addition, we included the mass as the input parameter (0.22 g). OptCom function was run with the generated community model as input, selecting 'original' as the strategy parameter and setting a min_growth of

0.01 h$^{-1}$ for each species. The community growth rate was maximized simultaneously with all individual growth rates, and fluxes were computed. The input fluxes were provided as environmental fluxes in mmol h$^{-1}$, and thus, they had to be divided by the total community biomass (0.22 g) to simulate growth and multiplied by the total community biomass to output again environmental fluxes.

**cFBA.** cFBA was run using Python 3.7 and COBRApy 0.24.0 following the methodology implemented in a previous computational study, where a community model of the same two species was built [79]. 'GLPK' was used as the default solver installed with COBRApy. The community model was built manually, unifying the extracellular metabolites and extracellular reactions common between species in one single reaction or metabolite. We used the same community model without adding the extra reactions described in the latter publication. The species biomass reactions were constrained to the relative abundance multiplied by the growth rate (g h$^{-1}$), and the same as the community growth rate (0.021 h$^{-1}$). The solution space was computed using the 'sample' function in the flux_analysis submodule found in COBRApy. The results shown here are the average and standard deviation based on 10000 samples generated under the specified condition.

**SteadyCom.** MATLAB 2022b was installed using an academic license, and COBRA Toolbox version 3.33 was installed following the installation instructions specified in the GitHub repository of COBRA Toolbox. SteadyCom was run using the 'SteadyCom' subroutine in COBRA Toolbox and 'GLPK', the LP solver installed when installing COBRA Toolbox. The community model was built using 'createMultipleSpeciesModel.' The names and IDs for metabolites and exchange reactions in the shared compartment of the community [u] were added into the community model as 'infoCom' and 'indCom' fields using 'getMultiSpeciesModelId.' The bounds of the reactions defined in the [u] compartment (e.g., 'autoIEX_MG[u]tr, 'EX_MG[u]') were imported from the Excel file 'Bounds_steadycom.xlsx'. SteadyCom was run using the built community model and the following options as input parameters: GRguess = 0.5; algorithm = 1; feasCrit = 1 and BMweight = 0.2. The community growth rate was maximized, and fluxes were computed.

**Microbiome Modelling Toolbox (MMT).** MATLAB 2022b was installed using an academic license, and COBRA Toolbox version 3.33 was installed following the installation instructions specified in the GitHub repository of COBRA Toolbox. MMT 2.0 was run from COBRA Toolbox using 'GLPK,' the LP solver installed when installing COBRA Toolbox. The community model was created using the 'joinModelsPairwiseFromList' function. The coupling factor 'c' and threshold 'u' were maintained at their default values, 400 and 0, respectively. The bounds of the reactions defined in the [u] compartment of the generated community model were imported and defined from the Excel file 'Bounds_MMT'. The species biomass reactions were constrained to the relative abundance multiplied by the growth rate (g h$^{-1}$). The interactions and fluxes of the co-culture were next explored using the function called 'simulatePairwiseInteractions'. To run the latter function, the community model and the pairedModelInfo.mat file created by 'joinModelsPairwiseFromList' were used as the input parameters. Besides, 'saveSolutionsFlag' was selected to output the fluxes.

**MICOM.** MICOM was run using Python 3.7 and COBRApy version 0.24.0. MICOM 0.32.2 was installed using pip as described in the GitHub repository of MICOM. 'OSQP' was installed by default with MICOM and was used as the solver. The community model was created using the 'Community' function following the procedure described to run OptCom. 'The cooperate tradeoff' algorithm was used with a fraction of 0.5 to simulate growth. The input fluxes were provided as environmental fluxes in mmol h$^{-1}$, and thus, they had to be divided by the total community biomass (0.22 g) to simulate growth, and multiplied by the total community biomass to output again environmental fluxes.

## Evaluation of the dynamic tools

Similar to what was done by Hanly and coworkers [73], *S. cerevisiae* S288C (iND750) [103] and *E. coli* K-12 substr. MG1655's (iJR904) [104] GEMs, acquired from the BiGG database [105], were used to perform the model simulations with DyMMM, DFBAlab, MMODES, and μbialSim. Furthermore, the *E. coli* model (iJR904) was modified by constraining flux bounds for glucose exchange and glucose kinase at zero to mathematically reflect the associated gene deletions. The simulations were based on an aerobic xylose co-culture of *S. cerevisiae* and the engineered *E. coli* strain ZSC113 fermentation experiment. In this experiment, glucose and xylose concentrations are expected to decrease over time, while the ethanol concentration is expected to increase. Simultaneously, S. cerevisiae and E. coli biomass concentrations are expected to increase over time. For all simulations, any constraints and parameters given in the original tools were modified according to the experimental values from the respective dataset from Hanly and coworkers [73]. Please see the S5, S6, S7 and S8 Tables for specific inputs as well as summary of boundary conditions, constraints, and parameters for all GEMS for the dynamic modeling approaches. The simulated growth and metabolite concentration curves of *E. coli* and *S. cerevisiae*, as well as from glucose, xylose, and ethanol were compared to the experimental measurements.

The coefficient of determination, also known as R-squared ($R^2$) was calculated to quantitatively estimate the quality of the model fits and performance with the experimental data as in the work of Montgomery [106] for the dynamic and spatiotemporal cases.

The equation for R-squared is:

$$R^2 = 1 - \frac{SSE}{SST}$$

where: $R^2$ is the coefficient of determination, $SSE$ is the sum of squared residuals, and $SST$ is the total sum of squares, and $\frac{SSE}{SST}$ represents the ratio of the sum of squared residuals to the total sum of squares.

The equation for the sum of squared residuals:

$$SSE = \sum_{i=1}^{n} (y_i - \hat{y}_i)^2$$

where: $SSE$ is the sum of squared residuals, $y_i$ is the actual value of the dependent variable for the $i^{th}$ observation, $\hat{y}_i$ is the predicted value of the dependent variable for the $i^{th}$ observation, and $n$ is the number of observations.

The equation for the total sum of squares:

$$SST = \sum_{i=1}^{n} (y_i - \bar{y})^2$$

where: $SST$ is the total sum of squares, $y_i$ is the actual value of the dependent variable for the $i^{th}$ observation, $\bar{y}$ is the mean of the dependent variable, and $n$ is the number of observations.

**dyMMM.** An implementation for this method is provided by the developers on sourceforge.net: https://sourceforge.net/p/dymmm/wiki/Home/. This method requires a working installation of COBRA Toolbox [107] version 3.33 was used in this study with MATLAB 2022b. The provided formulation allows users to add initial concentrations of limiting or important substrates as well as products of interest. There is also a field to add initial biomass concentrations of microbes. We followed as instructed in the source code by the developers to add the proper inputs regarding initial concentration values of substrates and microbial biomasses, as well as GEMs, where needed. These inputs we added as presented in the supplementary table (S6, S7 and S8 Tables).

**DFBAlab.** An implementation for this method is provided by the developers via the website https://yoric.mit.edu/software. This method requires a working installation of IBM CPLEX (version 20.1.0 was used in this study with MATLAB 2022b) or Gurobi (we used Gurobi version 10.0). A similar example is supplied in the tutorial to the user as the case study used in this work. Although the original formulation of this method uses values directly from the literature [62], in the implementation, the values are not exactly as reported for our chosen case study. Therefore, the initial concentrations of glucose, xylose, and microbial biomasses were adapted S7 and S8 Tables), and the time span of the simulation was modified from 10 to 14 hours.

**µbialSim.** An implementation for this method is supported via the website: https://github.com/fcentler/microbialSim. This method requires a working installation of COBRA Toolbox (version 3.33 was used in this study with MATLAB 2022b) or CellNetAnalyzer (version 2022.1 was used in this study). The provided example 2, based on a batch-culture growth of a binary syntrophic community, served as a starting point for employing our dynamic case study conditions. The reactor was defined according to initial substrate concentrations (see S8 Table). The GEMs (iJR904 and iND750) were loaded and adapted according to the COBRA Toolbox protocol within µbialSim. In addition, reaction indices for the non-growth associate maintenance (NGAM) and biomass reaction were provided. Exchange reactions were defined that link compounds in the reactor to cellular uptake and secretion. The models were parameterized according to the kinetic parameters provided in the literature (S7 and S8 Tables). The simulation duration and time step size chosen were 14 and 0.02 h, respectively. dFBA was selected as the solver type between the options ODE and dFBA. However, GPLK was used as the COBRA solver.

**MMODES.** An implementation for this method is supported via the website: https://mmodes.readthedocs.io/en/latest/index.html. MMODES was installed via pip (using Python 3.9 in this study as well as COBRApy version 0.24.0). The example script and the documentation were used for instructions. First, the medium file was defined according to the medium composition given and initial conditions from the work of Hanly and Henson (see S7 and S8 Tables). Next, the kinetic parameters were defined and set according to values used by Hanly and Henson (see S7 Table). Then, the GEMs (iJR904 and iND750) of the microbes, along with their respective initial biomasses and substrates they consume, were added to the model framework. The FBA optimization routine using the GPLK solver was selected. We chose a simulation time of 14 h with 0.02 h timesteps. Also, finite element analysis was used for the ordinary differential equation integrator.

## Evaluation of the spatiotemporal tools

Similar to what was done by Harcombe et al. [33], simulations were performed using the GEMs for *E. coli* (iJO1366) [108] and *S. enterica* (iRR1083) [109]. We compared BacArena, IndiMeSH, and CROMICS, which were designed to study microbial communities using FBA and IBMs, with COMETS which was mainly designed to investigate the interrelationships of bacterial communities in space using FBA and PLMs. The goal was to compare the methods with one another for their potential in simulating trophic dependences of multispecies bacterial communities without making prior assumptions. Therefore, simulations mimicked the two-member consortium experiment studied in the original publication [33]. This consortium was composed of two mutant strains, *S. enterica* LT2 and *E. coli* K-12. *S. enterica* LT2 cannot metabolize and consume lactose, and *E. coli* K-12 cannot produce methionine. Hence, these two species participate in a mutualistic relationship because *S. enterica* relies on the secretion of acetate for a substrate by *E. coli* while *E. coli* needs *S. enterica* to produce methionine.

Stoichiometric models of each species were modified to incorporate known genetic constraints according to the method of Harcombe and coworkers [33]. For instance, in the *E. coli* strain, the metB mutation was accounted for by constraining to zero the flux through the corresponding reaction (cystathionine $\gamma$-synthase). In *S. enterica*, methionine necessitated that we added a gain-of-function mutation in metA (homoserine transsuccinylase). This secretion was modeled as coupled to biomass where lactose is utilized by *E. coli*, so that as cells grew, they produced appropriate amino acid levels. A summary of boundary conditions, constraints, and parameters for all GEMS for spatiotemporal methods can be found in S10 and S11 Tables).

**COMETS.** The *in silico* experiment was set up as reported in the work of Harcombe et al. [33] (for more detail, see experimental procedures in the original work). The developers provide an implementation for this method via the website: https://www.runcomets.org/. This method requires a working installation of Java and Gurobi (we used Java 20 in this study as well as Gurobi version 10.0). The Python version of the COMETS Toolbox (we used Python version 3.9 in this study as well as COMETS v0.4.1) was installed using the Anaconda distribution.

**BacArena.** The developers provide a tutorial to help implement this method via https://bacarena.github.io/. This method requires a working installation of R and Sybil (we used R version 4.2.2 in this study as well as Sybil version 2.2.0). Both GEMs of *E. coli* (iJO1366) and *S. enterica* (STM v1 multistrain) were retrieved from the BiGG database [105]. The *S. enterica* model was modified to ensure methionine production using the method of Harcombe et al. [33]. The biomass reaction was updated to include the production of 0.5 mmol gDW$^{-1}$ of excreted extracellular methionine, which was balanced by an equal amount of intracellular methionine consumed. The methionine transporter (METtex, METabcpp) were set to export only (upper bounds set to 0). The *E. coli* model was modified to block flux through the corresponding reaction, cystathionine $\gamma$-synthase (CYSTL). To model metabolic exchanges between the microbes and compare the results of BacArena with the other methods, we performed the simulations with our method using a setup similar to COMETS [33]. The simulations were carried out on a 50 times 50 grid environment for 48 hours. In the setups, a minimal medium stated in Harcombe et al. [33] was added to the environment with 1 mmol of lactose per grid position, oxygen, and several co-factors (calcium, chlorine, cobalt, potassium, iron, magnesium, ammonium, manganese, nickel, phosphate, zinc, and sulfate). To ensure the growth and mimic the preculture environment, an initial amount of 1.0 mmol methionine and acetate was added to each grid position. The amounts and biomasses of the initial species were also set according to initial frequencies used in Harcombe et al. [33]. The diffusion of metabolites was calibrated to the standard diffusion of glucose.

**IndiMeSH.** The developers provide an implementation for this method in the supporting materials of the original paper [38]. This method requires a working installation of MATLAB (we used MATLAB version 2022b in this study). The *in silico* experiment was performed to simulate microbial life on a two-dimensional surface in IndiMeSH, a rectangular, saturated pore network (coordination number: 4) was created with the same topology as in the COMETS simulations (for details, see S1(A) Fig in the supplementary material of the original paper). Each pore has dimensions of 500x250x70 microns (LxWxH), as specified in the paper. Furthermore, we followed the setup as reported in the work of Borer et al. [38] (for details, see the Methods section in the original paper).

**CROMICS.** The developers provide an implementation for this method via a GitHub repository: https://github.com/EPFL-LCSB/cromics. This method requires a working installation of MATLAB and CPLEX solver (version 20.1.0 was used in this study with MATLAB 2022b). We used the setup detailed in the original paper [71], which mimics the setup of Harcombe and coworkers [33] for the two-species consortium of *S. enterica* and *E. coli* (for details, see the Methods section in the original paper [71]).

## Supporting information

**S1 Fig. Qualitative assessment of the static tools/approaches with the numerical information.**
(TIF)

**S2 Fig. Qualitative assessment of the dynamic tools with the numerical information.**
(TIF)

**S3 Fig. Qualitative assessment of the spatiotemporal tools with numerical information.**
(TIF)

**S1 Table. Summarized comparison of the available static tools/approaches.**
(PDF)

**S2 Table. Rubric—Qualitative assessment of tools/approaches.**
(PDF)

**S3 Table. Inputs, outputs, and assumptions of the static tools/approaches.**
(PDF)

**S4 Table. Genome-scale metabolic models (GEMs) and input parameters used as constraints in some static tools/approaches to model the co-culture of *C. autoethanogenum* and *C. kluyveri*.**
(PDF)

**S5 Table. Summarized comparison of the available dynamic tools/approaches.**
(PDF)

**S6 Table. Inputs, outputs, and assumptions of the dynamic tools/approaches.**
(PDF)

**S7 Table. Genome-scale metabolic models (GEMs), GEM constraints, substrate uptake parameters, and initial biomass concentrations used by some dynamic tools/approaches to model the co-culture of *S. cerevisiae* and *E. coli*.**
(PDF)

**S8 Table. Initial substrate concentrations used for the co-culture dynamic tools/approaches model the co-culture of *S. cerevisiae* and *E. coli*.**
(PDF)

**S9 Table. Summarized comparison of the available spatiotemporal tools/approaches.**
(PDF)

**S10 Table. Inputs, outputs, and assumptions of the spatiotemporal tools/approaches.**
(PDF)

**S11 Table. Genome-scale metabolic models (GEMs) and input parameters used as constraints in some spatiotemporal tools/approaches to model the co-culture of *S. enterica* and *E. coli*.**
(PDF)

**S1 Text. Brief description of the set of tools and approaches evaluated in this study.**
(DOCX)

**S1 Data. Reported $R^2$ values with simulation versus experimental data.**
(XLSX)

## Acknowledgments

We thank Dr. William R. Harcombe and Dr. Jeremy Chacon from the Department of Biological Sciences at the University of Minnesota for assistance with the COMETS Toolbox for spatiotemporal modeling and interpretation of some experimental and model results. We thank Dr. Costas D. Maranas and Dr. Mohammad M. Islam from the Pennsylvania State University Department of Chemical Engineering for providing the d-OptCom tool and making suggestions for how to use the software. We thank Dr. Louca Stilianos from the Institute of Ecology and Evolution at the University of Oregon for correspondence and assistance employing the Microbial Community Modeler (MCM) tool. We thank Dr. Christian Diener from the Gibbons Laboratory at the Institute of Systems Biology for assistance with MICOM.

## Author Contributions

**Conceptualization:** William T. Scott, Jr., Sara Benito-Vaquerizo, Maria Suarez-Diez, Peter J. Schaap.

**Data curation:** William T. Scott, Jr., Sara Benito-Vaquerizo.

**Formal analysis:** William T. Scott, Jr., Sara Benito-Vaquerizo, Johannes Zimmermann.

**Funding acquisition:** Maria Suarez-Diez, Peter J. Schaap.

**Investigation:** William T. Scott, Jr., Sara Benito-Vaquerizo.

**Methodology:** William T. Scott, Jr., Sara Benito-Vaquerizo, Maria Suarez-Diez.

**Resources:** Maria Suarez-Diez, Peter J. Schaap.

**Software:** William T. Scott, Jr., Johannes Zimmermann, Djordje Bajić, Almut Heinken.

**Supervision:** William T. Scott, Jr., Peter J. Schaap.

**Validation:** William T. Scott, Jr., Sara Benito-Vaquerizo.

**Visualization:** William T. Scott, Jr., Sara Benito-Vaquerizo.

**Writing – original draft:** William T. Scott, Jr., Sara Benito-Vaquerizo.

**Writing – review & editing:** William T. Scott, Jr., Sara Benito-Vaquerizo, Johannes Zimmermann, Djordje Bajić, Almut Heinken, Maria Suarez-Diez, Peter J. Schaap.

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
