## [Decision Letter · Decision Letter 0]

10 Apr 2023

Dear Dr. Scott,

Thank you very much for submitting your manuscript "A Structured Evaluation of Genome-Scale Constraint-Based Modeling Tools for Microbial Consortia" for consideration at PLOS Computational Biology. As with all papers reviewed by the journal, your manuscript was reviewed by members of the editorial board and by several independent reviewers. The reviewers appreciated the attention to an important topic. Based on the reviews, we are likely to accept this manuscript for publication, providing that you modify the manuscript according to the review recommendations.

The reviewers are positive about your manuscript that provides a comparative evaluation of tools for constraint based modeling of microbial communities. They, however, recommend some improvements including quantitative assesment of the tools and proper documentation.

Sincerely,

Tunahan Cakir

Guest Editor

PLOS Computational Biology

Kiran Patil

Section Editor

PLOS Computational Biology

The reviewers are positive about your manuscript that provides a comparative evaluation of tools for constraint based modeling of microbial communities. They, however, recommend some improvements including quantitative assesment of the tools and proper documentation.

Reviewer's Responses to Questions

**Comments to the Authors:**

Reviewer #1: Revision Plos Computational Biology

Scott Jr et all present a timely and complete review addressing the comparative analysis of available software for microbial communities modelling. The authors address the detailed analysis of steady-state, dynamic and spatio-temporal modelling tools. The authors rank the tools according their performance but also in the context of FAIR principles. The dataset of selected tools is very complete and very representative of the state of the field. There are some weaknesses limiting the scope of the review, I suggest to address some of they in order to increase the value of the current manuscript.

Major Comments

1. The review addresses the analysis of steady-state, dynamic and spatiotemporal tools, however there are interesting alternatives to analyse microbial communities such as SMETANA (Zelezniak et al 2014) and Community Gap-filling (Giannari et al 2021) which identify interspecific interactions or DOLMN (Thommes et al 2019) and FLYCOP (Garcia-Jimenez et al 2018) which address the synthetic designing of microbial consortia. In order to increase the scope of the review perhaps such methods could be reviewed and ranked as well.

2. To increase the understanding of the review I suggest to include potential software licenses associated with each method since it is an important aspect when selecting the software.

3. In the case of the assignment of flexibility is required a more detailed information from the authors. For me it’s not clear what mean satisfactory or good, for instance. Overall, I miss a more detailed description of the qualitative measures used to create the different scores used in the text. They look actually very subjective. There are multiple qualitative measures, but some of the variables can be objectively measured. For instance, the reproducibility or scalability the data can be quantitatively analysed against available data to corroborate the score.

4. Overall, I miss some software traceability in term of software version. The authors don’t mention the software versions used, or downloading date. This is important for the reproducibility of the analysis.

5. In discussion, the authors observed that the qualitative assessment correlate to the quantitative assessment, but they tested only the tools with highest qualitative score, so this assumption is somewhat weak. Can the authors extend the discussion of this topic?

6. The figures in general have low resolution and the text and numbers of the axis are too small.

7. The authors omit the version/year of publication of the tools in the figures 2,4 and 7. So the potential readers cannot identify the version of the software used. I suggest adding this information in the legend.

8. I miss the scripts used by the authors for the analysis following FAIR principles.

Minor Comments

1. Line 398 COBRAPy � COBRApy

2.

3. Agora 2.0 paper has been already published (https://www.nature.com/articles/s41587-022-01628-0 ) please use the new citation instead of the BioRxiv preprint.

Reviewer #2: I have enjoyed reading the manuscript entitled "A Structured Evaluation of Genome-Scale Constraint-Based

Modeling Tools for Microbial Consortia" and found it to be a valuable and timely contribution to the field. The authors have done an excellent job in systematically evaluating COBRA-based tools for microbial communities, which addresses an important gap in the literature. This work is expected to provide the scientific community with guidance on selecting appropriate tools for their specific applications and to help developers prioritise improvements in the future.

The manuscript is well-written and well-organised, with clear objectives and a comprehensive methodology. The qualitative assessment of the 24 published tools using the FAIR principles is informative, and the quantitative testing of a subset of 14 tools against experimental data from three different case studies is commendable. The authors' discussion on the differences in the mathematical formulation of the approaches and their relation to the results provides valuable insights and shows a deep understanding of the topic.

However, I would like to suggest an improvement that could further enhance the rigour of the quantitative assessment. The authors could consider incorporating additional quantitative Key Performance Indicators (KPIs), such as error and correlation of dynamic behaviours of the metabolic system. These KPIs can help to evaluate the accuracy of the predictions made by the COBRA-based tools, and allow for a more comprehensive comparison between them. By including these metrics, readers will have a better understanding of the strengths and limitations of each tool and be better equipped to make informed decisions on which tool to use for their specific research question. Also in discussion one could add that every tool can be as accurate as the model is and that the reader should pay take all quantitive evaluations with a pinch of salt... And most important, besides just providing a link to zenodo repository, I could not find a documentation notebook that puts all results together or explain how to install entire set of analyses, this is absolutely crucial for publication.

In conclusion, I believe that the manuscript is an important and well-executed study that will significantly contribute to the field of microbial community research. I recommend its acceptance for publication, with the suggested improvement to the quantitative assessment. This work will undoubtedly serve as a valuable resource for researchers and developers working with COBRA-based tools and microbial communities.

**Have the authors made all data and (if applicable) computational code underlying the findings in their manuscript fully available?**

Reviewer #1: **No: **There are some scripts not provided

Reviewer #2: **No: **The authors should add more documentation, perhaps installed as a package/container with all the analyses and have a detailed documentation of the reproducibility of results.

PLOS authors have the option to publish the peer review history of their article (what does this mean?). If published, this will include your full peer review and any attached files.

Reviewer #1: **Yes: **Juan Nogales

Reviewer #2: **Yes: **Aleksej Zelezniak

Figure Files:

Data Requirements:

Reproducibility:

References:

---

## [Decision Letter · Decision Letter 1]

7 Jun 2023

Dear Dr. Scott, Jr.,

Thank you very much for submitting the revised version of your manuscript "A structured evaluation of genome-scale constraint-based modeling tools for microbial consortia" for consideration at PLOS Computational Biology. Based on the reviews, we are likely to accept this manuscript for publication, providing that you modify the manuscript according to the review recommendations.

As recommended by one of the reviewers, the authors should reorganize the repository they share alongwith the manuscript to enable easy reproduction of the figures in the manuscript.

Sincerely,

Tunahan Cakir

Guest Editor

PLOS Computational Biology

Kiran Patil

Section Editor

PLOS Computational Biology

Reviewer's Responses to Questions

**Comments to the Authors:**

Reviewer #1: The authors have satisfactorily answered my previous concerns and the work is more than suitable for publication. I consider that the current manuscript is a valuable piece for the community interested in modeling and engineering microbiomes.

Reviewer #2: The repository authors provided is exceptionally challenging to navigate and understand. There are hundreds of files, this is unclear how to run them to get the figure the authors provide in the manuscript. Many methods are just shown as they are, with documentation from original papers, with tons of comment code. Thus it remains unclear how to reproduce the study (what are the input files for the figures, and how did you generate them?). I highly recommend that authors have another revision and seriously address this comment; this will improve the citation count of their work and provide a valuable resource for the community. There is too much irreproducible research out there, and this work should not be adding to that pile. I am not asking to containerise everything, but having a script showing how to plot individual figures and how the input files for figures were generated is essential (at least commenting on which tool was run).

**Have the authors made all data and (if applicable) computational code underlying the findings in their manuscript fully available?**

Reviewer #1: Yes

Reviewer #2: **No: **The repository authors provide is highly challenging to reproduce. Many methods are just shown as they are, with documentation from original papers. I doubt the authors can reproduce the study themselves.

PLOS authors have the option to publish the peer review history of their article (what does this mean?). If published, this will include your full peer review and any attached files.

Reviewer #1: **Yes: **Juan Nogales

Reviewer #2: **Yes: **Aleksej Zelezniak

Figure Files:

Data Requirements:

Reproducibility:

References:

---

## [Decision Letter · Decision Letter 2]

17 Jul 2023

Dear Dr. Scott, Jr.,

We are pleased to inform you that your manuscript 'A structured evaluation of genome-scale constraint-based modeling tools for microbial consortia' has been provisionally accepted for publication in PLOS Computational Biology.

Best regards,

Tunahan Cakir

Guest Editor

PLOS Computational Biology

Kiran Patil

Section Editor

PLOS Computational Biology

Reviewer's Responses to Questions

**Comments to the Authors:**

Reviewer #2: The authors adequately addressed previously raised concerns.

**Have the authors made all data and (if applicable) computational code underlying the findings in their manuscript fully available?**

Reviewer #2: Yes

PLOS authors have the option to publish the peer review history of their article (what does this mean?). If published, this will include your full peer review and any attached files.

Reviewer #2: **Yes: **Aleksej Zelezniak

---

## [Editor Report · Acceptance letter]

4 Aug 2023

PCOMPBIOL-D-23-00221R2 

A structured evaluation of genome-scale constraint-based modeling tools for microbial consortia

Dear Dr Scott, Jr.,

I am pleased to inform you that your manuscript has been formally accepted for publication in PLOS Computational Biology. Your manuscript is now with our production department and you will be notified of the publication date in due course.

With kind regards,

Zsofi Zombor
